# Towards Understanding the Condensation of Neural Networks at Initial Training

**Hanxu Zhou**[1]**, Qixuan Zhou**[1]**, Tao Luo**[1,2]**, Yaoyu Zhang**[1,3,]***, Zhi-Qin John Xu**[1,]†

[1] School of Mathematical Sciences, Institute of Natural Sciences, MOE-LSC
and Qing Yuan Research Institute, Shanghai Jiao Tong University
[2] CMA-Shanghai, Shanghai Artificial Intelligence Laboratory
[3] Shanghai Center for Brain Science and Brain-Inspired Technology

## Abstract

Empirical works show that for ReLU neural networks (NNs) with small initialization, input weights of hidden neurons (the input weight of a hidden neuron consists of the weight from its input layer to the hidden neuron and its bias term) condense onto isolated orientations. The condensation dynamics implies that the training implicitly regularizes a NN towards one with much smaller effective size. In this work, we illustrate the formation of the condensation in multi-layer fully connected NNs and show that the maximal number of condensed orientations in the initial training stage is twice the multiplicity of the activation function, where "multiplicity" indicates the multiple roots of activation function at origin. Our theoretical analysis confirms experiments for two cases, one is for the activation function of multiplicity one with arbitrary dimension input, which contains many common activation functions, and the other is for the layer with one-dimensional input and arbitrary multiplicity. This work makes a step towards understanding how small initialization leads NNs to condensation at the initial training stage.

## 1 Introduction

The question why over-parameterized neural networks (NNs) often show good generalization attracts much attention (Breiman, 1995; Zhang et al., 2021). Luo et al. (2021) found that when initialization is small, the input weights of hidden neurons in two-layer ReLU NNs (the input weight or the feature of a hidden neuron consists of the weight from its input layer to the hidden neuron and its bias term) condense onto isolated orientations during the training. As illustrated in the cartoon example in Fig. 1, the condensation transforms a large network, which is often over-parameterized, to the one of only a few effective neurons, leading to an output function with low complexity. Since, in most cases, the complexity bounds the generalization error (Bartlett and Mendelson, 2002), the study of condensation could provide insight to how over-parameterized NNs are implicitly regularized to achieve good generalization performance in practice.

Small initialization leads NNs to rich non-linearity during the training (Mei et al., 2019; Rotskoff and Vanden-Eijnden, 2018; Chizat and Bach, 2018; Sirignano and Spiliopoulos, 2020). For example, in over-parameterized regime, small initialization can achieve low generalization error (Advani et al., 2020). Irrespective of network width, small initialization can make two-layer ReLU NNs converge to a solution with maximum margin (Phuong and Lampert, 2020). Small initialization also enables neural networks to learn features actively (Lyu et al., 2021; Luo et al., 2021). The gradual increment of the condensed orientations is consistent with many previous works, which show that the network

---

*Corresponding author: zhyy.sjtu@sjtu.edu.cn.
†Corresponding author: xuzhiqin@sjtu.edu.cn.

output evolves from simple to complex during the training process (Xu et al., 2020; Rahaman et al., 2019; Arpit et al., 2017). The initial condensation resets the network to a simple state, which brings out the simple to complex training process. Condensation is an important phenomenon that reflects the feature learning process. Therefore, it is important to understand how condensation emerges during the training with small initialization.

**Illustration of Condensation**

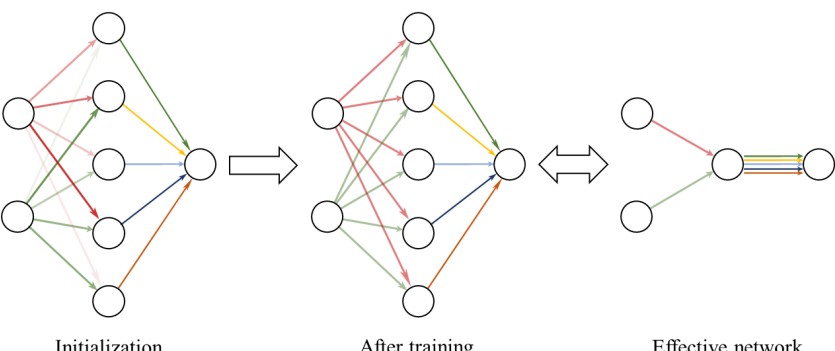

Initialization     After training     Effective network

Figure 1: Illustration of condensation. The color and its intensity of a line indicate the strength of the weight. Initially, weights are random. Soon after training, the weights from an input node to all hidden neurons are the same, i.e., condensation. Multiple hidden neurons can be replaced by an effective neuron with low complexity, which has the same input weight as original hidden neurons and the output weight as the summation of all output weights of original hidden neurons.

The dynamic behavior of the training at the initial state is important for the whole training process, because it largely determines the training dynamics of a neural network and the region it ends up in (Fort et al., 2020; Hu et al., 2020), which impacts the characteristics of the neural networks in the final stage of training (Luo et al., 2021; Jiang et al., 2019; Li et al., 2018). For two-layer ReLU NNs, several works have studied the mechanism underlying the condensation at the initial training stage when the initialization of parameters goes to zero (Maennel et al., 2018; Pellegrini and Biroli, 2020). However, it still remains unclear that **for NNs of more general activation functions, how the condensation emerges at the initial training stage.**

In this work, we show that the condensation at the initial stage is closely related to the multiplicity $p$ at $x = 0$, which means the derivative of activation at $x = 0$ is zero up to the $(p-1)th$-order and is non-zero for the $p$-th order. Many common activation functions, e.g., $\tanh(x)$, $\mathrm{sigmoid}(x)$, $\mathrm{softplus}(x)$, $\mathrm{Gelu}(x)$, $\mathrm{Swich}(x)$, etc, are all multiplicity $p = 1$, and $x\tanh(x)$ and $x^2\tanh(x)$ have multiplicity two and three, respectively. Our contribution is summarized as follows:

- Our extensive experiments suggest that the maximal number of condensed orientations in the initial training is twice the multiplicity of the activation function used in general NNs.
- We present a theory for the initial condensation with small initialization for two cases, one is for the activation function of multiplicity one with arbitrary dimension input, and the other is for the layer with one-dimensional input and arbitrary multiplicity. As many common activation functions are multiplicity $p = 1$, our theory would be of interest to general readers.

## 2 Related works

Luo et al. (2021) systematically study the effect of initialization for two-layer ReLU NN with infinite width by establishing a phase diagram, which shows three distinct regimes, i.e., linear regime (similar to the lazy regime) (Jacot et al., 2018; Arora et al., 2019; Zhang et al., 2020; E et al., 2020; Chizat and Bach, 2019), critical regime (Mei et al., 2019; Rotskoff and Vanden-Eijnden, 2018; Chizat and Bach, 2018; Sirignano and Spiliopoulos, 2020) and condensed regime (non-linear regime), based on

the relative change of input weights as the width approaches infinity, which tends to 0, $O(1)$ and $+\infty$, respectively. Zhou et al. (2022) make a step towards drawing a phase diagram for three-layer ReLU NNs with infinite width, revealing the possibility of completely different dynamics (linear, critical and condensed dynamics) emerging within a deep NN for its different layers. As shown in Luo et al. (2021), two-layer ReLU NNs with infinite width do not condense in the neural tangent kernel (NTK) regime , slightly condense in the mean-field regime, and clearly condense in the non-linear regime. However, in Luo et al. (2021), it is not clear how general the condensation phenomenon is when other activation functions are used and why there is condensation.

Zhang et al. (2021a,b); Fukumizu and Amari (2000); Fukumizu et al. (2019); Simsek et al. (2021) propose a general Embedding Principle of loss landscape of DNNs that unravels a hierarchical structure of the loss landscape of NNs, i.e., loss landscape of an DNN contains all critical points of all the narrower DNNs. The embedding principle shows that a large DNN can experience critical points where the DNN condenses and its output is the same as that of a much smaller DNN. However, the embedding principle does not explain how the training can take the DNN to such critical points.

The condensation is consistent with previous works that suggest that NNs may learn data from simple to complex patterns (Arpit et al., 2017; Xu et al., 2019; Rahaman et al., 2019; Xu et al., 2020; Jin et al., 2020; Kalimeris et al., 2019). For example, an implicit bias of frequency principle is widely observed that NNs often learn the target function from low to high frequency (Xu et al., 2019; Rahaman et al., 2019; Xu et al., 2020), which has been utilized to understand various phenomena (Ma et al., 2020; Xu and Zhou, 2021) and inspired algorithm design (Liu et al., 2020; Cai et al., 2020; Tancik et al., 2020; Li et al., 2020, 2021).

# 3 Preliminary: Neural networks and initial stage

A two-layer NN is

$$f_{\boldsymbol{\theta}}(\boldsymbol{x}) = \sum_{j=1}^{m} a_j \sigma(\boldsymbol{w}_j \cdot \boldsymbol{x}), \tag{1}$$

where $\sigma(\cdot)$ is the activation function, $\boldsymbol{w}_j = (\bar{\boldsymbol{w}}_j, \boldsymbol{b}_j) \in \mathbb{R}^{d+1}$ is the neuron feature including the input weight and bias terms, and $\boldsymbol{x} = (\bar{\boldsymbol{x}}, 1) \in \mathbb{R}^{d+1}$ is the concatenation of the input sample and scalar 1, $\boldsymbol{\theta}$ is the set of all parameters, i.e., $\{a_j, \boldsymbol{w}_j\}_{j=1}^{m}$. For simplicity, **we call $\boldsymbol{w}_j$ input weight or weight** and $\boldsymbol{x}$ input sample.

A $L$-layer NN can be recursively defined by setting the output of the previous layer as the input to the current hidden layer, i.e.,

$$\boldsymbol{x}^{[0]} = (\boldsymbol{x}, 1), \quad \boldsymbol{x}^{[1]} = (\sigma(\boldsymbol{W}^{[1]}\boldsymbol{x}^{[0]}), 1), \quad \boldsymbol{x}^{[l]} = (\sigma(\boldsymbol{W}^{[l]}\boldsymbol{x}^{[l-1]}), 1), \text{ for } l \in \{2, 3, ..., L\}$$
$$f(\boldsymbol{\theta}, \boldsymbol{x}) = \boldsymbol{a}^{\mathsf{T}}\boldsymbol{x}^{[L]} \triangleq f_{\boldsymbol{\theta}}(\boldsymbol{x}), \tag{2}$$

where $\boldsymbol{W}^{[l]} = (\bar{\boldsymbol{W}}^{[l]}, \boldsymbol{b}^{[l]}) \in \mathbb{R}^{m_l \times (m_{l-1}+1)}$, and $m_l$ represents the dimension of the $l$-th hidden layer. For simplicity, **we also call each row of $\boldsymbol{W}^{[l]}$ input weight or weight** and $\boldsymbol{x}^{[l-1]}$ input to the $l$-th hidden layer. The target function is denoted as $f^*(\boldsymbol{x})$. The training loss function is the mean squared error

$$R(\boldsymbol{\theta}) = \frac{1}{2n} \sum_{i=1}^{n} (f_{\boldsymbol{\theta}}(\boldsymbol{x}_i) - f^*(\boldsymbol{x}_i))^2. \tag{3}$$

Without loss of generality, we assume that the output is one-dimensional for theoretical analysis, because, for high-dimensional cases, we only need to sum up the components directly. For summation, it does not affect the results of our theories. We consider the gradient flow training

$$\dot{\boldsymbol{\theta}} = -\nabla_{\boldsymbol{\theta}} R(\boldsymbol{\theta}). \tag{4}$$

To ensure that the training is close to the gradient flow, all the learning rates used in this paper are sufficiently small. We characterize the activation function by the following definition.

*Definition* 1 (multiplicity $p$). Suppose that $\sigma(x)$ satisfies the following condition, there exists a $p \in \mathbb{N}^*$, such that the $s$-th order derivative $\sigma^{(s)}(0) = 0$ for $s = 1, 2, \cdots, p-1$, and $\sigma^{(p)}(0) \neq 0$, then we say $\sigma$ has multiplicity $p$.

*Remark* 3.1. Here are some examples. $\tanh(x)$, $\mathrm{sigmoid}(x)$ and $\mathrm{softplus}(x)$ have multiplicity $p = 1$. $x\tanh(x)$ has multiplicity $p = 2$.

We illustrate small initialization and initial stage as follows

**Small initialization**: $\boldsymbol{W}^{[l]} \sim o(1)$ and $\boldsymbol{W}^{[l]}\boldsymbol{x}^{[l-1]} \sim o(1)$ for all $l$'s and $\boldsymbol{x}^{[l-1]}$.

We want to remark that it dose not make sense to define the initial stage by the number of training steps, because the training is affected by many factors, such as the learning rate. Therefore, in our experiments, the epochs we use to show phenomena can cover a wide range. Alternatively, we consider the initial stage as follows.

**Initial stage**: the period when the leading-order Taylor expansion w.r.t activated neurons is still valid for theoretical analysis of Sec. 5.

As the parameters of a neural network evolve, the loss value will also decay accordingly, which is easy to be directly observed. Therefore, we propose *an intuitive definition of the initial stage* of training by the size of loss in this article, that is the stage before the value of loss function decays to 70% of its initial value. Such a definition is reasonable, for generally a loss could decay to 1% of its initial value or even lower. The loss of the all experiments in the article can be found in Appendix A.3, and they do meet the definition of the initial stage here.

**Cosine similarity:** The cosine similarity between two vectors $\boldsymbol{u}$ and $\boldsymbol{v}$ is defined as

$$D(\boldsymbol{u}, \boldsymbol{v}) = \frac{\boldsymbol{u}^{\mathsf{T}}\boldsymbol{v}}{(\boldsymbol{u}^{\mathsf{T}}\boldsymbol{u})^{1/2}(\boldsymbol{v}^{\mathsf{T}}\boldsymbol{v})^{1/2}}. \tag{5}$$

## 4   Initial condensation of input weights

In this section, we would empirically show how the condensation differs among NNs with activation function of different multiplicities in the order of a practical example, multidimensional synthetic data, and 1-d input synthetic data, followed by theoretical analysis in the next section.

### 4.1   Experimental setup

For synthetic dataset: Throughout this work, we use fully-connected neural network with size, $d$-$m$-$\cdots$-$m$-$d_{\mathrm{out}}$. The input dimension $d$ is determined by the training data. The output dimension is $d_{\mathrm{out}} = 1$. The number of hidden neurons $m$ is specified in each experiment. All parameters are initialized by a Gaussian distribution $N(0, \mathrm{var})$. The total data size is $n$. The training method is Adam with full batch, learning rate $\mathrm{lr}$ and MSE loss. We sample the training data uniformly from a sub-domain of $\mathbb{R}^d$. *The sampling range and the target function are chosen randomly to show the generality of our experiments.*

For CIFAR10 dataset: We use Resnet18-like neural network, which has been described in Fig. 2, and the input dimension is $d = 32 \times 32 \times 3$. The output dimension is $d_{\mathrm{out}} = 10$. All parameters are initialized by a Gaussian distribution $N(0, \mathrm{var})$. The total data size is $n$. The training method is Adam with batch size 128, learning rate $\mathrm{lr}$ and cross-entropy loss.

### 4.2   A practical example

The condensation of the weights of between the fully-connected (FC) layers of a Resnet18-like neural network on CIFAR10 is shown in Fig. 2, whose activation functions for FC layers are $\tanh(x)$, $\mathrm{sigmoid}(x)$, $\mathrm{softplus}(x)$ and $x\tanh(x)$, indicated by the corresponding sub-captions, respectively. As shown in Fig. 2(a), for activation function $\tanh(x)$, the color indicates cosine similarity $D(\boldsymbol{u}, \boldsymbol{v})$ of two hidden neurons' weights, whose indexes are indicated by the abscissa and the ordinate, respectively. If the neurons are in the same beige block, $D(\boldsymbol{u}, \boldsymbol{v}) \sim 1$ (navy-blue block, $D(\boldsymbol{u}, \boldsymbol{v}) \sim -1$), their input weights have the same (opposite) direction. Input weights of hidden neurons in Fig. 2(a) condense at two opposite directions, i.e., one line. Similarly, weights of hidden neurons for NNs with $\mathrm{sigmoid}(x)$ and $\mathrm{softplus}(x)$ (Fig. 2(b, c)), which are frequently used and have multiplicity one, condense at one direction. As the multiplicity increases, NNs with $x\tanh x$ (Fig. 2(d)) condense at two different lines. These experiments suggest that the condensation is closely related to the multiplicity of the activation function.

In these experiments, we find that the performance of the Resnet18-like network with tanh FC layers with small initialization is similar to the one with common initialization in Appendix A.4.

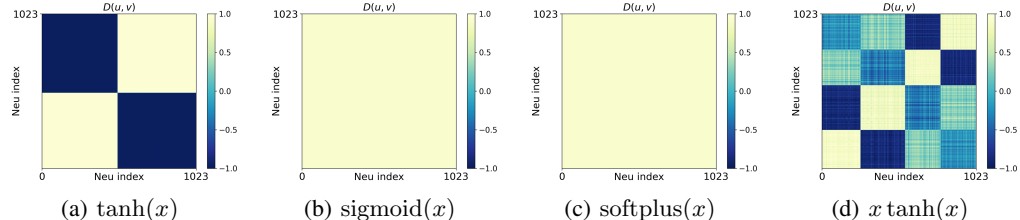

(a) $\tanh(x)$      (b) $\mathrm{sigmoid}(x)$      (c) $\mathrm{softplus}(x)$      (d) $x\tanh(x)$

Figure 2: Condensation of Resnet18-like neural networks on CIFAR10. Each network consists of the convolution part of resnet18 and fully-connected (FC) layers with size 1024-1024-10 and softmax. The color in figures indicates the cosine similarity of normalized input weights of two neurons in the first FC layer, whose indexes are indicated by the abscissa and the ordinate, respectively. The convolution part is equipped with ReLU activation and initialized by Glorot normal distribution (Glorot and Bengio, 2010). The activation functions are $\tanh(x)$, $\mathrm{sigmoid}(x)$, $\mathrm{softplus}(x)$ and $x\tanh(x)$ for FC layers in (a), (b), (c), and (d), separately. The numbers of steps selected in the sub-pictures are epoch 20, epoch 30, epoch 30 and epoch 61, respectively. The learning rate is $3 \times 10^{-8}, 1 \times 10^{-8}, 1 \times 10^{-8}$ and $5 \times 10^{-6}$, separately .The FC layers are initialized by $N(0, \frac{1}{m_{\mathrm{out}}^3})$, and Adam optimizer with cross-entropy loss and batch size 128 are used for all experiments.

### 4.3 Multidimensional synthetic data

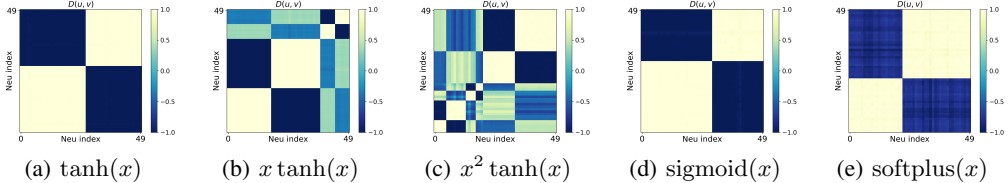

(a) $\tanh(x)$     (b) $x\tanh(x)$     (c) $x^2\tanh(x)$     (d) $\mathrm{sigmoid}(x)$     (e) $\mathrm{softplus}(x)$

Figure 3: Condensation of two-layer NNs. The color indicates $D(\boldsymbol{u}, \boldsymbol{v})$ of two hidden neurons' input weights at epoch 100, whose indexes are indicated by the abscissa and the ordinate, respectively. If neurons are in the same beige block, $D(\boldsymbol{u}, \boldsymbol{v}) \sim 1$ (navy-blue block, $D(\boldsymbol{u}, \boldsymbol{v}) \sim -1$), their input weights have the same (opposite) direction. The activation functions are indicated by the sub-captions. The training data is 80 points sampled from $\sum_{k=1}^{5} 3.5\sin(5x_k + 1)$, where each $x_k$ is uniformly sampled from $[-4, 2]$. $n = 80$, $d = 5$, $m = 50$, $d_{\mathrm{out}} = 1$, var $= 0.005^2$. lr $= 10^{-3}, 8 \times 10^{-4}, 2.5 \times 10^{-4}$ for (a-c), (d) and (e), respectively.

For convenience of experiments, we then use synthetic data to perform extensive experiments to study the relation between the condensation and the multiplicity of the activation function.

We use two-layer fully-connected NNs with size 5-50-1 to fit $n = 80$ training data sampled from a 5-dimensional function $\sum_{k=1}^{5} 3.5\sin(5x_k + 1)$, where $\boldsymbol{x} = (x_1, x_2, \cdots, x_5)^{\mathsf{T}} \in \mathbb{R}^5$ and each $x_k$ is uniformly sampled from $[-4, 2]$. As shown in Fig. 3(a), for activation function $\tanh(x)$, input weights of hidden neurons condense at two opposite directions, i.e., one line. As the multiplicity increases, NNs with $x\tanh(x)$ (Fig. 3(b)) and $x^2\tanh x$ (Fig. 3(c)) condense at two and three different lines, respectively. For activation function $\mathrm{sigmoid}(x)$ in Fig. 3(d) and $\mathrm{softplus}(x)$ in Fig. 3(e), NNs also condense at two opposite directions.

For multi-layer NNs with different activation functions, we show that the condensation for all hidden layers is similar to the two-layer NNs. In deep networks, residual connection is often introduced to overcome the vanishing of gradient. To show the generality of condensation, we perform an experiment of six-layer NNs with residual connections. To show the difference of various activation functions, we set the activation functions for hidden layer 1 to hidden layer 5 as $x^2\tanh(x)$, $x\tanh(x)$, $\mathrm{sigmoid}(x)$, $\tanh(x)$ and $\mathrm{softplus}(x)$, respectively. The structure of the residual is

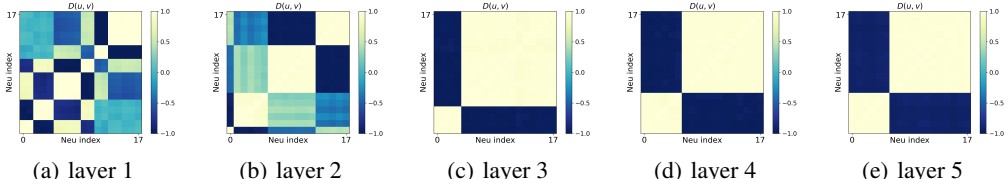

(a) layer 1     (b) layer 2     (c) layer 3     (d) layer 4     (e) layer 5

Figure 4: Condensation of six-layer NNs with residual connections. The activation functions for hidden layer 1 to hidden layer 5 are $x^2 \tanh(x)$, $x \tanh(x)$, $\mathrm{sigmoid}(x)$, $\tanh(x)$ and $\mathrm{softplus}(x)$, respectively. The numbers of steps selected in the sub-pictures are epoch 1000, epoch 900, epoch 900, epoch 1400 and epoch 1400, respectively, while the NN is only trained once. The color indicates $D(\boldsymbol{u}, \boldsymbol{v})$ of two hidden neurons' input weights, whose indexes are indicated by the abscissa and the ordinate, respectively. The training data is 80 points sampled from a 3-dimensional function $\sum_{k=1}^{3} 4 \sin(12x_k + 1)$, where each $x_k$ is uniformly sampled from $[-4, 2]$. $n = 80$, $d = 3$, $m = 18$, $d_{\mathrm{out}} = 1$, var $= 0.01^2$, lr $= 4 \times 10^{-5}$.

$\boldsymbol{h}_{l+1}(\boldsymbol{x}) = \sigma(\boldsymbol{W}_l \boldsymbol{h}_l(\boldsymbol{x}) + \boldsymbol{b}_l) + \boldsymbol{h}_l(\boldsymbol{x})$, where $\boldsymbol{h}_l(\boldsymbol{x})$ is the output of the $l$-th layer. As shown in Fig. 4, input weights condense at three, two, one, one and one lines for hidden layer 1 to hidden layer 5, respectively. Note that residual connections are not necessary. We show an experiment of the same structure as in Fig. 4 but without residual connections in Appendix A.5.

Through these experiments, we conjecture that the maximal number of condensed orientations at initial training is twice the multiplicity of the activation function used. To understand the mechanism of the initial condensation, we turn to experiments of 1-d input and two-layer NNs, which can be clearly visualized in the next subsection.

### 4.4   1-d input and two-layer NN

For 1-d data, we visualize the evolution of the two-layer NN output and each weight, which confirms the connection between the condensation and the multiplicity of the activation function.

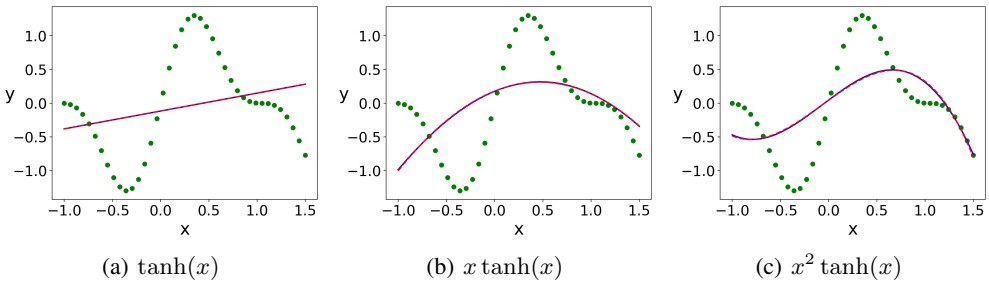

(a) $\tanh(x)$     (b) $x \tanh(x)$     (c) $x^2 \tanh(x)$

Figure 5: The outputs of two-layer NNs at epoch 1000 with activation function $\tanh(x)$, $x \tanh(x)$, and $x^2 \tanh(x)$ are displayed, respectively. The training data is 40 points uniformly sampled from $\sin(3x) + \sin(6x)/2$ with $x \in [-1, 1.5]$, illustrated by green dots. The blue solid lines are the NN outputs at test points, while the red dashed auxiliary lines are the first, second, third and first order polynomial fittings of the test points for (a, b, c), respectively. Parameters are $n = 40$, $d = 1$, $m = 100$, $d_{\mathrm{out}} = 1$, var $= 0.005^2$, lr $= 5 \times 10^{-4}$.

We display the outputs at initial training in Fig. 5. Due to the small magnitude of parameters, an activation function with multiplicity $p$ can be well approximated by a $p$-th order polynominal, thus, the NN output can also be approximated by a $p$-th order polynominal. As shown in Fig. 5, the NN outputs with activation function $\tanh(x)$, $x \tanh(x)$ and $x^2 \tanh(x)$ overlap well with the auxiliary of a linear, a quadratic and a cubic polynominal curve, respectively in the beginning. This experiment, although simple, but convincingly shows that NN does not always learn a linear function at the initial training stage and the complexity of such learning depends on the activation function.

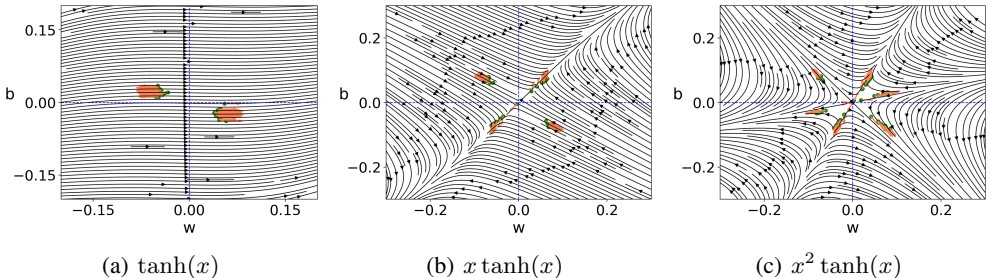

(a) $\tanh(x)$  (b) $x\tanh(x)$  (c) $x^2\tanh(x)$

Figure 6: The direction field for input weight $\boldsymbol{w} := (w, b)$ of the dynamics in (4.4) at epoch 200. All settings are the same as Fig. 5. Around the original point, the field has one, two, three stables lines, on which an input weight would keep its direction, for $\tanh(x)$, $x\tanh(x)$, and $x^2\tanh(x)$, respectively. We also display the value of each weight by the green dots and the corresponding directions by the orange arrows.

We visualize the direction field for input weight $\boldsymbol{w}_j := (w_j, b_j)$, following the gradient flow,

$$\dot{\boldsymbol{w}}_j = -\frac{a_j}{n} \sum_{i=1}^{n} e_i \sigma'(\boldsymbol{w}_j \cdot \boldsymbol{x}_i) \boldsymbol{x}_i,$$

where $e_i := f_{\boldsymbol{\theta}}(\boldsymbol{x}_i) - f^*(\boldsymbol{x}_i)$. Since we only care about the direction of $\boldsymbol{w}_j$ and $a_j$ is a scalar at each epoch, we can visualize $\dot{\boldsymbol{w}}_j$ by $\dot{\boldsymbol{w}}_j/a_j$. For simplicity, we do not distinguish $\dot{\boldsymbol{w}}_j/a_j$ and $\dot{\boldsymbol{w}}_j$ if there is no ambiguity. When we compute $\dot{\boldsymbol{w}}_j$ for different $j$'s, $e_i \boldsymbol{x}_i$ for $(i = 1, \cdots, n)$ is independent with $j$. Then, at each epoch, for a set of $\{e_i, \boldsymbol{x}_i\}_{i=1}^{n}$, we can consider the following direction field

$$\dot{\boldsymbol{\omega}} = -\frac{1}{n} \sum_{i=1}^{n} e_i \boldsymbol{x}_i \sigma'(\boldsymbol{\omega} \cdot \boldsymbol{x}_i).$$

When $\boldsymbol{\omega}$ is set as $\boldsymbol{w}_j$, we can obtain $\dot{\boldsymbol{w}}_j$. As shown in Fig. 6, around the original point, the field has one, two, three stables lines, on which a neuron would keep its direction, for $\tanh(x)$, $x\tanh(x)$, and $x^2\tanh(x)$, respectively. We also display the input weight of each neuron on the field by the green dots and their corresponding velocity directions by the orange arrows. Similarly to the high-dimensional cases, NNs with multiplicity $p$ activation functions condense at $p$ different lines for $p = 1, 2, 3$. Therefore, It is reasonable to conjecture that the maximal number of condensed orientations is twice the multiplicity of the activation function used.

Taken together, we have empirically shown that the multiplicity of the activation function is a key factor that determines the complexity of the initial output and condensation. To facilitate the understanding of the evolution of condensation in the initial stage, we show several steps during the initial stage of each example in Appendix A.6.

## 5 Analysis of the initial condensation of input weights

In this section, we would present a preliminary analysis to understand how the multiplicity of the activation function affects the initial condensation. At each training step, we consider the velocity field of weights in each hidden layer of a neural networks.

Considering a network with $L$ hidden layers, we use row vector $\boldsymbol{W}_j^{[k]}$ to represent the weight from the $(k-1)$-th layer to the $j$-th neuron in the $k$-th layer. Since condensation is always accompany with small initialization, together with the initial stage defined in Sec. 3, we make the following assumptions,

*Assumption* 1. Small initialization infers that $\boldsymbol{W}_j^{[k]} \boldsymbol{x}^{[k-1]} \sim o(1)$ applies for all $k$'s and $j$'s.

*Assumption* 2. During the initial stage of condensation, Taylor expansion of each corresponding activated neurons holds, i.e.,

$$\sigma'(\boldsymbol{W}_j^{[k]} \boldsymbol{x}^{[k-1]}) = \sigma'(0) + \cdots + \frac{\sigma^{(\gamma)}(0)}{(\gamma-1)!} (\boldsymbol{W}_j^{[k]} \boldsymbol{x}^{[k-1]})^{\gamma-1} + O((\boldsymbol{W}_j^{[k]} \boldsymbol{x}^{[k-1]})^{\gamma}) \qquad (6)$$

Suppose the activation function has multiplicity $p$, i.e., $\sigma^{(s)}(0) = 0$ for $s = 1, 2, \cdots, p-1$, and $\sigma^{(p)}(0) \neq 0$. Then, together with Assumption 2, we have

$$\sigma'(\boldsymbol{W}_j^{[k]} \boldsymbol{x}^{[k-1]}) \approx \frac{\sigma^{(p)}(0)}{(p-1)!} (\boldsymbol{W}_j^{[k]} \boldsymbol{x}^{[k-1]})^{p-1}. \tag{7}$$

For each $k$ and $j$, $\boldsymbol{W}_j^{[k]}$ satisfies the following dynamics, (see Appendix A.2)

$$\dot{r} = \boldsymbol{u} \cdot \dot{\boldsymbol{w}}, \quad \dot{\boldsymbol{u}} = \frac{\dot{\boldsymbol{w}} - (\dot{\boldsymbol{w}} \cdot \boldsymbol{u})\boldsymbol{u}}{r}. \tag{8}$$

where $\boldsymbol{w}$ can represent $\boldsymbol{W}_j^{[k]\mathsf{T}}$ for all $k$'s and $j$'s, $r = \|\boldsymbol{w}\|_2$ is the amplitude, and $\boldsymbol{u} = \boldsymbol{w}/r$.

For convenience, we define an operator $\mathcal{P}$ satisfying $\mathcal{P}\boldsymbol{w} := \dot{\boldsymbol{w}} - \boldsymbol{u}(\dot{\boldsymbol{w}} \cdot \boldsymbol{u})$. To specify the condensation for theoretical analysis, we make the following definition,

**Condensation**: the weight evolves towards a direction which will not change in the direction field and is defined as follows,

$$\dot{\boldsymbol{u}} = 0 \iff \mathcal{P}\boldsymbol{w} := \dot{\boldsymbol{w}} - \boldsymbol{u}(\dot{\boldsymbol{w}} \cdot \boldsymbol{u}) = 0. \tag{9}$$

Since $\dot{\boldsymbol{w}} \cdot \boldsymbol{u}$ is a scalar, $\dot{\boldsymbol{w}}$ is parallel with $\boldsymbol{u}$. $\boldsymbol{u}$ is a unit vector, therefore, we have $\boldsymbol{u} = \pm \dot{\boldsymbol{w}}/\|\dot{\boldsymbol{w}}\|_2$.
*Remark* 1 (An intuitive explanation for condensation). In this work, we consider NNs with sufficiently small parameters. Suppose $r = \|\boldsymbol{w}\|_2 \sim O(\epsilon)$, where $\epsilon$ is a small quantity, then dynamics (8) will show that $O(\dot{r}) \sim O(\dot{\boldsymbol{w}})$ and $O(\dot{\boldsymbol{u}}) \sim O(\dot{r})/O(\epsilon)$. Here $O(\dot{r}) \sim O(\dot{\boldsymbol{w}})$ refers that the evolution of $\dot{r}$ is at the same order as every component of $\dot{\boldsymbol{w}}$, and it is the same for $O(\dot{\boldsymbol{u}}) \sim O(\dot{r})/O(\epsilon)$. Therefore, the orientation $\boldsymbol{u}$ would vary much more quickly than the amplitude $r$. By the dynamics for $\boldsymbol{w}$ (Eq. 10) and the Taylor approximation with multiplicity $p$ (Eq. 7), it is easy to find that the solutions for Eq. 9 are finite. Then, taken together, the orientation $\boldsymbol{u}$ would converge rapidly into certain directions, leading to condensation.

In the following, we study the case of (i) $p = 1$ and (ii) $m_{k-1} = 1$ (the dimension of input of the $k$-th layer equals one), and reach the following informal proposition,

*Proposition* 1. Suppose that Assumption 1 and 2 holds. Consider the leading-order Taylor expansion of Eq. 9, as initialization towards zero. If either $p = 1$ or $m_{k-1} = 1$ holds, then the maximal number of roots for Eq. 9 is twice of the multiplicity of the activation function.

*Proof.* **Case 1:** $p = 1$

By gradient flow, we can obtain the dynamics for $\boldsymbol{W}_j^{[k]}$ (see Appendix A.2),

$$\dot{\boldsymbol{w}}^{\mathsf{T}} = \dot{\boldsymbol{W}}_j^{[k]} = -\frac{1}{n} \sum_{i=1}^n (f(\boldsymbol{\theta}, \boldsymbol{x}_i) - y_i) \left[ \mathrm{diag}\{\sigma'(\boldsymbol{W}^{[k]} \boldsymbol{x}_i^{[k-1]})\} (\boldsymbol{E}^{[k+1:L]} \boldsymbol{a}) \right]_j \boldsymbol{x}_i^{[k-1]\mathsf{T}}, \tag{10}$$

where we use $\boldsymbol{E}^l = \boldsymbol{W}^{[l]\mathsf{T}} \mathrm{diag}\{\sigma'(\boldsymbol{W}^{[l]} \cdot \boldsymbol{x}^{[l-1]})\}$, for $l \in \{2, 3, ..., L\}$, $\boldsymbol{E}^{[q:p]} = \boldsymbol{E}^q \boldsymbol{E}^{q+1} ... \boldsymbol{E}^p$, and $\boldsymbol{x}_i^{[k]}$ represents the neurons of the $k$-th layer generated by the $i$-th sample.

For a fixed step, we only consider the gradient of loss w.r.t. $\boldsymbol{W}_j^{[k]}$. According to our assumption $p = 1$, we have $\sigma'(0) \neq 0$. Suppose that parameters are small and denote $e_i := f(\boldsymbol{\theta}, \boldsymbol{x}_i) - y_i$. By Taylor expansion,

$$\mathcal{P}\boldsymbol{w} \stackrel{\text{leading order}}{\approx} \mathcal{Q}\boldsymbol{w} := -\frac{1}{n}\{(\mathrm{diag}\{\sigma'(\boldsymbol{0})\}(\boldsymbol{E}^{[k+1:L]} \boldsymbol{a}))_j \cdot \sum_{i=1}^n e_i \boldsymbol{x}_i^{[k-1]}\}$$

$$+\{(\frac{1}{n}(\mathrm{diag}\{\sigma'(\boldsymbol{0})\}(\boldsymbol{E}^{[k+1:L]} \boldsymbol{a}))_j \cdot \sum_{i=1}^n e_i \boldsymbol{x}_i^{[k-1]} \cdot \boldsymbol{u})\boldsymbol{u}\} = 0,$$

where operator $\mathcal{Q}$ is the leading-order approximation of operator $\mathcal{P}$, and here $\boldsymbol{E}^{[k+1:L]}$ is independent with $i$, because $\text{diag}\{\sigma'(\boldsymbol{W}^{[l]}\boldsymbol{x}^{[l-1]})\} \approx \text{diag}\{\sigma'(0)\}$. Since $\text{diag}\{\sigma'(\boldsymbol{0})\} = c\boldsymbol{I}, c \neq 0$ by assumption, and, WLOG, we assume $\boldsymbol{a} \neq 0$, then

$$\mathcal{Q}\boldsymbol{w} = 0 \iff \sum_{i=1}^{n} e_i\boldsymbol{x}_i^{[k-1]} = \left(\sum_{i=1}^{n} e_i\boldsymbol{x}_i^{[k-1]} \cdot \boldsymbol{u}\right)\boldsymbol{u}.$$

We have

$$\boldsymbol{u} = \frac{\sum_{i=1}^{n} e_i\boldsymbol{x}_i^{[k-1]}}{\|\sum_{i=1}^{n} e_i\boldsymbol{x}_i^{[k-1]}\|_2} \quad or \quad \boldsymbol{u} = -\frac{\sum_{i=1}^{n} e_i\boldsymbol{x}_i^{[k-1]}}{\|\sum_{i=1}^{n} e_i\boldsymbol{x}_i^{[k-1]}\|_2}.$$

This calculation shows that for layer $k$, the input weights for any hidden neuron $j$ have the same two stable directions. Together with the analysis before, i.e., when parameters are sufficiently small, the orientation $\boldsymbol{u}$ would move much more quickly than the amplitude $r$, all input weights would move towards the same direction or the opposite direction, i.e., condensation on a line, under small initialization.

**Case 2: the $k$-th layer with one-dimensional input, i.e., $m_{k-1} = 1$**

By the definition of the multiplicity $p$, we have

$$\sigma'(\boldsymbol{w} \cdot \boldsymbol{x}_i) = \frac{\sigma^{(p)}(0)}{(p-1)!}(\boldsymbol{w} \cdot \boldsymbol{x}_i)^{p-1} + o((\boldsymbol{w} \cdot \boldsymbol{x}_i)^{p-1}).$$

where $(\cdot)^{p-1}$ and $\sigma^{(p)}(\cdot)$ operate on component here. Then up to the leading order in terms of the magnitude of $\boldsymbol{\theta}$, we have (see Appendix A.2)

$$\mathcal{P}\boldsymbol{w} \overset{\text{leading order}}{\approx} \mathcal{Q}\boldsymbol{w} := -\{(\frac{1}{n}\sum_{i=1}^{n} e_i\boldsymbol{x}_i^{[k-1]}(\boldsymbol{w}^{\mathsf{T}}\boldsymbol{x}_i^{[k-1]})^{p-1}) \cdot [\text{diag}\{\frac{\sigma^{(p)}(\boldsymbol{0})}{(p-1)!}\}(\boldsymbol{E}^{[k+1:L]}\boldsymbol{a})]_j\}$$

$$+\{((\frac{1}{n}\sum_{i=1}^{n} e_i\boldsymbol{x}_i^{[k-1]}(\boldsymbol{w}^{\mathsf{T}}\boldsymbol{x}_i^{[k-1]})^{p-1}) \cdot [\text{diag}\{\frac{\sigma^{(p)}(\boldsymbol{0})}{(p-1)!}\}(\boldsymbol{E}^{[k+1:L]}\boldsymbol{a})]_j \cdot \boldsymbol{u})\boldsymbol{u}\}.$$

WLOG, we also assume $\boldsymbol{a} \neq 0$. And by definition, $\boldsymbol{w} = r\boldsymbol{u}$, we have

$$\mathcal{Q}\boldsymbol{w} = 0 \iff \boldsymbol{u} = \frac{\frac{1}{n}\sum_{i=1}^{n} e_i\boldsymbol{x}_i^{[k-1]}(\boldsymbol{u}^{\mathsf{T}}\boldsymbol{x}_i^{[k-1]})^{p-1}}{\|\frac{1}{n}\sum_{i=1}^{n} e_i\boldsymbol{x}_i^{[k-1]}(\boldsymbol{u}^{\mathsf{T}}\boldsymbol{x}_i^{[k-1]})^{p-1}\|_2} \quad or \quad \boldsymbol{u} = -\frac{\frac{1}{n}\sum_{i=1}^{n} e_i\boldsymbol{x}_i^{[k-1]}(\boldsymbol{u}^{\mathsf{T}}\boldsymbol{x}_i^{[k-1]})^{p-1}}{\|\frac{1}{n}\sum_{i=1}^{n} e_i\boldsymbol{x}_i^{[k-1]}(\boldsymbol{u}^{\mathsf{T}}\boldsymbol{x}_i^{[k-1]})^{p-1}\|_2}.$$

Since $m_{k-1} + 1 = 2$ (for $\boldsymbol{x}^{[k-1]} = (\sigma(\boldsymbol{W}^{[k-1]}\boldsymbol{x}^{[k-2]}), 1)$ and $m_{k-1} = 1$ ), we denote $\boldsymbol{u} = (u_1, u_2)^{\mathsf{T}} \in \mathbb{R}^2$ and $\boldsymbol{x}_i^{[k-1]} = ((\boldsymbol{x}_i^{[k-1]})_1, (\boldsymbol{x}_i^{[k-1]})_2)^{\mathsf{T}} \in \mathbb{R}^2$, then,

$$\frac{\sum_{i=1}^{n}(u_1(\boldsymbol{x}_i^{[k-1]})_1 + u_2(\boldsymbol{x}_i^{[k-1]})_2)^{p-1}e_i(\boldsymbol{x}_i^{[k-1]})_1}{\sum_{i=1}^{n}(u_1(\boldsymbol{x}_i^{[k-1]})_1 + u_2(\boldsymbol{x}_i^{[k-1]})_2)^{p-1}e_i(\boldsymbol{x}_i^{[k-1]})_2} = \frac{u_1}{u_2} \triangleq \hat{u}.$$

We obtain the equation for $\hat{u}$,

$$\sum_{i=1}^{n}(\hat{u}(\boldsymbol{x}_i^{[k-1]})_1 + (\boldsymbol{x}_i^{[k-1]})_2)^{p-1}e_i(\boldsymbol{x}_i^{[k-1]})_1 = \hat{u}\sum_{i=1}^{n}(\hat{u}(\boldsymbol{x}_i^{[k-1]})_1 + (\boldsymbol{x}_i^{[k-1]})_2)^{p-1}e_i(\boldsymbol{x}_i^{[k-1]})_2.$$

Since it is an univariate $p$-th order equation, $\hat{u} = \frac{u_1}{u_2}$ has at most $p$ complex roots. Because $\boldsymbol{u}$ is a unit vector, $\boldsymbol{u}$ at most has $p$ pairs of values, in which each pair are opposite. $\qquad \square$

Taken together, our theoretical analysis is consistent with our experiments, that is, the maximal number of condensed orientations is twice the multiplicity of the activation function used when parameters are small. As many commonly used activation functions are either multiplicity $p = 1$ or ReLU-like, our theoretical analysis is widely applied and sheds light on practical training.

## 6   Discussion

In this work, we have shown that the characteristic of the activation function, i.e., multiplicity, is a key factor to understanding the complexity of NN output and the weight condensation at initial training. The condensation restricts the NN to be effectively low-capacity at the initial training stage, even for finite-width NNs. In the initial stage, the Taylor expansion upto the leading order indicates that the activation function can be approximated by a homogeneous polynomial. Thus, even the amplitudes of weights within a layer are not identical, the network can be reduced approximately. During the training, the NN increases its capacity to better fit the data, leading to a potential explanation for their good generalization in practical problems. This work also serves as a starting point for further studying the condensation for multiple-layer neural networks throughout the training process.

For general multiplicity with high-dimensional input data, the theoretical analysis for the initial condensation is a very difficult problem, which is equivalent to counting the number of the roots of a high-order multivariate polynomial with a special structure originated from NNs. Training data can also affect the condensation but not the maximal number of condensed orientations. When data is simple, such as low frequency, the number of the condensed orientations can be less, some experiments of MNIST and CIFAR100 can be found in Appendix A.7.

## Acknowledgments and Disclosure of Funding

This work is sponsored by the National Key R&D Program of China Grant No. 2022YFA1008200 (Z. X., T. L., Y. Z.), the Shanghai Sailing Program, the Natural Science Foundation of Shanghai Grant No. 20ZR1429000 (Z. X.), the National Natural Science Foundation of China Grant No. 62002221 (Z. X.), the National Natural Science Foundation of China Grant No. 12101401 (T. L.), Shanghai Municipal Science and Technology Key Project No. 22JC1401500 (T. L.), the National Natural Science Foundation of China Grant No. 12101402 (Y. Z.), Shanghai Municipal of Science and Technology Project Grant No. 20JC1419500 (Y.Z.), the Lingang Laboratory Grant No.LG-QS-202202-08 (Y.Z.), Shanghai Municipal of Science and Technology Major Project No. 2021SHZDZX0102, and the HPC of School of Mathematical Sciences and the Student Innovation Center, and the Siyuan-1 cluster supported by the Center for High Performance Computing at Shanghai Jiao Tong University, Key Laboratory of Marine Intelligent Equipment and System, Ministry of Education, P.R. China.

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
