# OpenReview forum: "Towards Understanding the Condensation of Neural Networks at Initial Training"
_NeurIPS.cc/2022/Conference — NeurIPS 2022 Accept_

### Official Review · Reviewer_Mpot · 2022-07-11

**Rating:** 6
**Confidence:** 3
**Soundness:** 3 good
**Presentation:** 3 good
**Contribution:** 2 fair

**Summary:**

The paper studies the condensation phenomenon qualitatively in neural networks with various activation functions. The number of condensed directions can be at most twice the multiplicity of the activation function, which is shown empirically and some theoretical calculations are given to back up.



**Questions:**

Can the authors comment on the difference between Figures 2 and 4 for the sigmoidal and softmax regarding the number of condensed directions? tanh is an odd function, so it makes sense to get $u$ and $-u$, but it is less clear for the other activation functions.

line 64: missing citations! The so-called embedding principle is identified first by Fukumizu and Amari [1] for the addition of one neuron, then generalized in Fukumizu et al. [2] and even more the construction and scaling of such manifolds of critical points is studied in Simsek et al. [3] from the perspective of permutation-symmetry.

line 101: dose -> does

line 103: "can across" unclear what is meant in this sentence

line 184: "It is reasonable" -> "it is reasonable"

line 220: "suppose we only consider the leading term" -> this is in no way a theorem statement, please remove the theorem format from the calculations.

I'm willing to increase my score if my concerns are addressed.

[1] Fukumizu, Kenji, and Shun-ichi Amari. "Local minima and plateaus in hierarchical structures of multilayer perceptrons." Neural networks 13.3 (2000): 317-327.

[2] Fukumizu, Kenji, et al. "Semi-flat minima and saddle points by embedding neural networks to overparameterization." Advances in neural information processing systems 32 (2019).

[3] Şimşek, Berfin, et al. "Geometry of the Loss Landscape in Overparameterized Neural Networks: Symmetries and Invariances." arXiv preprint arXiv:2105.12221 (2021).

**Limitations:**

Yes.

**Strengths And Weaknesses:**

Originality: As indicated by the authors, condensation is analyzed before for example by Maennel et al. The paper generalizes the analysis for multiple activation functions.

Quality: Various experiments are presented to study the phenomenon in models from ResNets to two-layer models, to a six-layer network with skip connections. The problem is with Theorem 5.1, although the calculations are informative, they should not be called theorems as they are not rigorous.

Clarity: Writing is coherent, but some individual sentences have slight English issues. The figures are nice and illustrative.

Significance: I think it is interesting for the NeurIps community to understand the initial training dynamics in neural networks. This can yield new inductive biases and training algorithms; also relevant for pruning, compressed networks etc.

---

> ### Author Response · Authors · 2022-08-01
> **Response to Reviewer Mpot**
>
> $\mathrm{Point 1: }$
>
> The problem is with Theorem 5.1, although the calculations are informative, they should not be called theorems as they are not rigorous.
>
> $\mathrm{Reply: }$
>
> Thank the reviewer for pointing out this inappropriate statement. We now replace 'Theorem' by 'Informal Proposition'.
>
> $\mathrm{Point 2: }$
>
> Can the authors comment on the difference between Figures 2 and 4 for the sigmoidal and softmax regarding the number of condensed directions? tanh is an odd function, so it makes sense to get -u and u, but it is less clear for the other activation functions.
>
> $\mathrm{Reply: }$
>
> The key difference between Figures 2 and 4 for the sigmoidal and softplus is that there is only one direction in Figure 2. First, we want to point out that the sigmoidal and softplus are both multiplicity 1. Therefore, the number of their condensed orientations should be less than or equal to 2, where the experiments are consistent with Theorem 5.1. Second, the key reason why u and -u make no difference is that (page 8) in the equation to obtain stable directions Qw=0, there are two u's that will cancel the effect of the sign of u. So whether the activation function is odd or not does not matter. Third, in Fig. 2, there are many convolutional layers before the fully-connected layers, which complicates the situation. It is not clear why in such a complicated case there is only one direction. Our current study is also looking into these phenomena.
>
> $\mathrm{Point 3: }$
>
> line 64: missing citations! The so-called embedding principle is identified first by Fukumizu and Amari [1] for the addition of one neuron, then generalized in Fukumizu et al. [2] and even more the construction and scaling of such manifolds of critical points is studied in Simsek et al. [3] from the perspective of permutation-symmetry.
>
> $\mathrm{Reply: }$
>
> Thank the reviewer for pointing out these references. We have added these in the manuscript.
>
> $\mathrm{Point 4: }$
>
> line 103: "can across" unclear what is meant in this sentence
>
> $\mathrm{Reply: }$
>
> Now, we revise "can cross" instead of "can across"
>
> $\mathrm{Point 5: }$
>
> line 220: "suppose we only consider the leading term" -> this is in no way a theorem statement, please remove the theorem format from the calculations.
>
> $\mathrm{Reply: }$
>
> Thank the reviewer for pointing out this inappropriate statement. We now rewrite the statement and replace 'Theorem' with 'Informal Proposition'.

---

### Official Review · Reviewer_qmpQ · 2022-07-11

**Rating:** 5
**Confidence:** 4
**Soundness:** 2 fair
**Presentation:** 3 good
**Contribution:** 2 fair

**Summary:**

Understanding how the weights of a neural network converge during training is an important task in deep learning. In this paper, the authors empirically investigate this problem for deep neural networks at the initial stage of training. Through a set of experiments on both synthetic and realistic data sets, they show that for small enough initialization, the orientation of the weight vectors to hidden neurons converge in a maximal number of directions as given by twice the multiplicity of the activation function. Here, the multiplicity is defined as the integer $p$ for which the derivative of the activation function at $x=0$ is zero up to the order $(p-1)$, and is non-zero for the $p$-th order.

Then, they provide some theoretical support for their experiments by presenting a preliminary analysis, in which they consider a Taylor approximation of the activation function at zero and show that, in the limit of vanishing initialization, convergence of orientations at early stage of training basically implies the alignment along a fixed number of directions as specified by the multiplicity. This analysis is done for activation functions with multiplicity 1 and high dimensional data, and for general multiplicity with 1 dimensional data.

**Questions:**

Comments and questions:
- There is some gap between the theoretical and empirical evidence: the theoretical analysis of the paper is done for gradient flow, but all the experiments seem to be done with Adam. How do the results look like if you use gradient descent (full batch size) with a small learning rate?
- It is mentioned in the introduction that condensation of orientation implies networks with small complexity. I agree that if the hidden neurons have identical directions then one can use the homogeneity of ReLU to rewrite the output and obtain a narrower network. However, this seems not possible for smooth activations. Thus, what are exactly the implications of condensation studied in your paper? What kind of generalization bounds can be proved for condensed weights?
- If my understanding is correct, the proof of Theorem 5.1 requires an implicit assumption, that is, the weight orientation must converge, and the convergence must occur near the initialization so that all the weights are still very small and the Taylor approximation is valid. However, I feel that this assumption is quite strong. It is not even clear a priori why the direction of the weight vectors should converge for this setting.

**Limitations:**

Limitations are discussed above. I do not see any societal impact.

**Strengths And Weaknesses:**

**Originality**
Previous work [Luo et al 2022] empirically observe that for a two-layer ReLU network with small initialization, the weights of hidden neurons condense on certain directions. However, it was not studied why it happens, and how it behaves for other activation functions. This paper fills this gap by studying the phenomenon for deep nets with smooth activation, and providing some intuitive explanation for why it happens.

**Quality**
The paper is easy to understand. However, there is not much discussion on the motivation and significance of their finding. For example, why it is important to study vanishing initialization if it is not even used in practice, how the behavior of the network at the initial stage of training could be informative for generalization where we only care about the final solution (the later stage of training).

**Clarity**
The writing is mostly clear.

**Significance**
I find the connection between the maximal number of condensed orientations and the multiplicity of activation interesting. I hope that this observation could be helpful for future researches in understanding implicit bias of gradient descent in the early stage of training.

---

> ### Author Response · Authors · 2022-08-01
> **Response to Reviewer qmpQ**
>
> $\mathrm{Point 1: }$
>
> The paper is easy to understand. However, there is not much discussion on the motivation and significance of their finding. For example, why it is important to study vanishing initialization if it is not even used in practice, how the behavior of the network at the initial stage of training could be informative for generalization where we only care about the final solution (the later stage of training).
>
> $\mathrm{Reply: }$
>
> In the second paragraph of the introduction, we have listed a series of literature to show small initialization leads NNs to rich non-linearity during the training. We would further add more discussion on the motivation of small initialization. The great advantage of using NNs is to utilize their non-linearity in learning compared with linear models. In the limit of vanishing initialization, many phenomena can be much clearer and easier studied, such as condensation. Taking a limit w.r.t. to some quantities is a common and useful approach in research. To study the over-parameterize regime of NNs, many works consider the two-layer network with infinite width [1]. To study the gradient descent, many works first study the gradient flow where the learning rate is infinitesimal [1]. To study the effect of random batch in training, many works study the batch size of 1 [2]. All these works consider the idealized settings but provide valuable insights in the deep learning field.
>
> The dynamic behavior of the training at the initial state is important for the whole training process, because it largely determines the training dynamics of a neural network and the region it ends up in (Third paragraph in Introduction). We would add more discussion about the motivation as follows. The gradual increment of the condensed orientations is consistent with many previous works, which shows that the network output evolves from simple to complex during the training process [3,4,5]. The initial condensation resets the network to a simple state, which brings out the simple-to-complex training process.
>
> [1] https://arxiv.org/abs/1806.07572
>
> [2] https://arxiv.org/abs/2101.12176
>
> [3] https://arxiv.org/abs/1806.08734
>
> [4] https://arxiv.org/abs/1901.06523
>
> [5] https://arxiv.org/abs/1706.05394
>
> $\mathrm{Point 2: }$
>
> There is some gap between the theoretical and empirical evidence: the theoretical analysis of the paper is done for gradient flow, but all the experiments seem to be done with Adam. How do the results look like if you use gradient descent (full batch size) with a small learning rate?
>
> $\mathrm{Reply: }$
>
> The initial condensation and the final condensation can be easily seen in gradient descent (full batch size) with a small learning rate. We have done many experiments with such a setting. Also the previous work [Luo et al. 2022] found the condensation in this setting. We choose Adam in the experimental part of the paper mainly because Adam algorithm is more common and has more practical significance in practice.

---

> ### Author Response · Authors · 2022-08-01
> **Response to Reviewer qmpQ**
>
> $\mathrm{Point 3: }$
>
> It is mentioned in the introduction that condensation of orientation implies networks with small complexity. I agree that if the hidden neurons have identical directions, then one can use the homogeneity of ReLU to rewrite the output and obtain a narrower network. However, this seems not possible for smooth activations. Thus, what are exactly the implications of condensation studied in your paper? What kind of generalization bounds can be proved for condensed weights?
>
> $\mathrm{Reply: }$
>
> For smooth activation function, as the reviewer points out, identical direction is not enough to reduce the network. We will clarify this point in the revised manuscript. Fig. 1 also requires identical weights to reduce the network. In Fig. 6, our results show that the weights have not only the same directions but also very approximate amplitudes after some training epochs. Thanks to the Taylor expansion, the activation function can be approximated by a homogeneous polynomial up to the leading order. Thus even the amplitudes are not identical, the network can be ``condensed'', at least approximately. For example, consider an activation approximated by $x^2$, the effective network uses the same input weight and uses an output weight equal to the sum of the square of amplitudes of all input weights.  In future work, we will study more quantitatively along this direction on the condensation phenomenon.
>
> $\mathrm{Point 4: }$
>
> If my understanding is correct, the proof of Theorem 5.1 requires an implicit assumption, that is, the weight orientation must converge, and the convergence must occur near the initialization so that all the weights are still very small and the Taylor approximation is valid. However, I feel that this assumption is quite strong. It is not even clear a priori why the direction of the weight vectors should converge for this setting.
>
> $\mathrm{Reply: }$
>
> In theorem 5.1, we only study the maximal number of roots for Eq. 9. This does not need the assumption of convergence. Our experiments support the convergence phenomena.

---

> > ### Comment · Reviewer_qmpQ · 2022-08-07
> > **Regarding convergence of weight orientation**
> >
> > Thank you for the response. However, I am confused by your answer regarding Point4. Specifically, if the weight orientation u(t) does not converge, then the LHS of (9) is never satisfied (i.e. $\dot{u}=0$).

---

> > > ### Author Response · Authors · 2022-08-09
> > > **We only study the number of steady solution, convergence is supported by experiments**
> > >
> > > Thanks for your response !
> > > We agree with the statement 'If the weight orientation $u(t)$ does not converge, then the L.H.S. of (9) would hardly be satisfied (i.e. $\dot{u}=0$) in practical'. And we indeed do not know whether the weight orientation $u(t)$ will converge or not theoretically in our paper. We will explore this further in our future studies. However, we want to clarify the following points.
> > > First, in theorem 5.1 (i.e. Informal proposition 1), we only consider the steady-state solution for Eq. 9, which is irrelevant with the convergence of the weight orientation $u(t)$.
> > > Second, in lines 214 to 222, we qualitatively analyze that the orientation $u$ would move much more quickly than than the amplitude $r$, and would converge rapidly into certain directions, leading to condensation.
> > > Third, as shown in experiments in Sec. 4, we empirically find that
> > > the weight orientation $u$ would condense into certain directions under small initialization at initial stage, which means that the weight orientation $u$ converges.
> > > Last, although we have not proved the convergence of the weight orientation $u$ theoretically yet, we will focus on it in our future work.

---

### Official Review · Reviewer_vTes · 2022-07-15

**Rating:** 5
**Confidence:** 4
**Soundness:** 3 good
**Presentation:** 3 good
**Contribution:** 1 poor

**Summary:**

The authors studied the empirical phenomenon of condensation during NN training, where the weight vectors of individual hidden neurons in a two-hidden-layer network concentrate in a few directions. The authors demonstrated empirically that the number of concentrated directions is related to the neuronal nonlinearity and specifically its multiplicity. The author then supplied a theoretical analysis under some heuristic assumptions of how this relationship arises.

**Questions:**

I would appreciate if the authors can better elucidate how the very condensed stage that the authors study is related to expressive, fully trained NNs.

**Limitations:**

See above.

**Strengths And Weaknesses:**

Strength: the writing is very clear. Concepts, assumptions and definitions are all well-defined, and the empirical and theoretical claims are clearly stated.
The empirical results connecting the number of condensed directions and the multiplicity of the nonlinearity seem very convincing, at least in the settings that the authors explored.

Weakness: My main critique is that I'm not sure that the studied regime is an interesting one. If I am interpreting Fig.2 correctly, the authors are suggesting that when using the tanh nonlinearity, the network condenses to an effective network with just two hidden neurons in this initial stage. The trained networks in practical settings will need much bigger effective sizes to fit training sets, and it is not clear to me how behaviors in this early, condensed regime are related to generalization and other phenomena of interest later on in training. The authors appear to suggest that condensation is important because condensation = smaller effective networks = fewer parameters = less overfitting. But first of all fewer parameters does not equate less overfitting, given what we now understand about various double descent phenomena. Second, while the fully trained networks may have effective sizes smaller than their real sizes, the effective size is almost certainly not as small as 2, as suggested by the tanh analysis.

---

> ### Author Response · Authors · 2022-08-01
> **Response to Reviewer vTes**
>
> $\mathrm{Point 1: }$
>
> My main critique is that I'm not sure that the studied regime is an interesting one. If I am interpreting Fig.2 correctly, the authors are suggesting that when using the tanh nonlinearity, the network condenses to an effective network with just two hidden neurons in this initial stage. The trained networks in practical settings will need much bigger effective sizes to fit training sets, and it is not clear to me how behaviors in this early, condensed regime are related to generalization and other phenomena of interest later on in training.
>
> The authors appear to suggest that condensation is important because condensation = smaller effective networks = fewer parameters = less overfitting. But first of all, fewer parameters does not equate less overfitting, given what we now understand about various double descent phenomena. Second, while the fully trained networks may have effective sizes smaller than their real sizes, the effective size is almost certainly not as small as 2, as suggested by the tanh analysis.
>
>  I would appreciate it if the authors could better elucidate how the very condensed stage that the authors study is related to expressive, fully trained NNs.
>
> $\mathrm{Reply: Part  \ 1}$
>
> We agree with the reviewer that "The trained networks in practical settings will need much bigger effective sizes to fit training sets". In the linear regime (e.g., NTK regime), the parameters of each neuron in the neural network would stay close to the corresponding initial values, i.e., no condensation happens. In such case, the number of effective neurons in the network can be very large.
> A one-dimensional example in [1] shows that the output is very oscillating with high complexity in the linear regime (Fig. 10 (a, e) in [1]). However, when the dropout is used during the training, condensation emerges and the output is much smoother (Fig. 10 (b, f) in [1]) with low complexity. Usually, a low-complexity function can generalize better. Even though the example in [1] is on a synthetic dataset, it requires more than two neurons to well fit the data.
>
> In practice, the final condensation is not prescribed but achieved through an optimization process. We aim to study how the GD optimization process achieves final condensation one-step-by-another.
>
> In this work, we study the initial training stage both empirically and theoretically. The initial condensation plays an important role for the later stage of training. The initial condensation resets the neural network to a simple state, such as two effective neurons in tanh network. Since the network in such a simple state is yet to well fit the training data, the number of condensed orientations would gradually increase during the training (see experiments in [5]). Although the training process is not fully understood, the gradual increment of the condensed orientations is consistent with many previous works, which show that the network output evolves from simple to complex during the training process [2,3,4]. Therefore, it is worth noticing that the effective size of 2 in tanh analysis is only an initial stage of the whole training process and it resets the network to a simple state, which brings out the simple-to-complex training process. In this sense, the initial condensation is an important basis for the later training stage. As suggested by the example in [1], condensation can affect the generalization, but a detailed study about the relation between (initial) condensation and generalization is left to the future.
>
> We also want to emphasize that the effective network is similar to a small network in the sense of expressivity, but different in optimization processes. Several previous works [5,6,7,8] have shown that when a small network at a critical point of the loss landscape is embedded into a large network (a kind of inverse process of condensation), the large network condenses and is also at a critical point. But the number of descent directions of the large network is usually much larger than the small one, which makes the training much easier. Compared to its effective small network, the large network with condensation is easier for training.

---

> ### Author Response · Authors · 2022-08-01
> **Response to Reviewer vTes**
>
> $\mathrm{Point 1: }$
>
> My main critique is that I'm not sure that the studied regime is an interesting one. If I am interpreting Fig.2 correctly, the authors are suggesting that when using the tanh nonlinearity, the network condenses to an effective network with just two hidden neurons in this initial stage. The trained networks in practical settings will need much bigger effective sizes to fit training sets, and it is not clear to me how behaviors in this early, condensed regime are related to generalization and other phenomena of interest later on in training.
>
> The authors appear to suggest that condensation is important because condensation = smaller effective networks = fewer parameters = less overfitting. But first of all, fewer parameters does not equate less overfitting, given what we now understand about various double descent phenomena. Second, while the fully trained networks may have effective sizes smaller than their real sizes, the effective size is almost certainly not as small as 2, as suggested by the tanh analysis.
>
>  I would appreciate it if the authors could better elucidate how the very condensed stage that the authors study is related to expressive, fully trained NNs.
>
> $\mathrm{Reply: Part  \ 2}$
>
> Finally, we want to clarify "condensation = smaller effective networks = fewer parameters = less overfitting" and "double descent". Fewer parameters do not equal to lower complexity. In our paper, we claim that few effective neurons lead to an output function with low complexity, instead of few parameters. Note that "fewer parameters = less overfitting" is a conventional wisdom. First, "double descent" happens in under-parameterized regime [9]. If the used network is in under-parameterized regime, there still exists condensation in the initial training stage, however, to fit the data as best as it can, the effective size of the network should be equal to its real size (no condensation). Second, in the over-parameterized and condensed regime, the number of condensed orientations would gradually increase during the training until the network can fit the training dataset. Although the number of effective parameters is small compared with the original big one, the effective small network is still out of the under-parameterized regime after training, i.e. in the over-parameterized regime. Therefore, the condensation of a large network would not make the large network generalize badly at final training stage. Third, the increasing number of condensed orientations leads to the increasing of the effective size. According to the double descent phenomenon w.r.t. to network size, it is possible to see the "double descent" of test error during the training process. [9] also reports such epoch-wise double descent.
>
> In summary, we thank the reviewer for pointing out this comment and we would limit our discussion of generalization within the over-parameterized regime and emphasize the low-complexity of condensation rather than few parameters. We revise the manuscript accordingly.
>
> [1] https://arxiv.org/abs/2207.05952
>
> [2] https://arxiv.org/abs/1706.05394
>
> [3] https://arxiv.org/abs/1806.08734
>
> [4] https://arxiv.org/abs/1901.06523
>
> [5] https://arxiv.org/abs/2105.14573
>
> [6] Fukumizu and Amari, "Local minima and plateaus in hierarchical structures of multilayer perceptrons." Neural networks 13.3 (2000).
>
> [7] https://arxiv.org/abs/1906.04868
>
> [8] https://arxiv.org/abs/2105.12221
>
> [9] https://arxiv.org/abs/1912.02292

---

> ### Author Response · Authors · 2022-08-09
> **Looking forward to hearing more**
>
> Dear Reviewer vTes,
>    Thanks for your comments, and we are looking forward to hearing more from you on our response.
> Best,
> Authors.

---

> ### Comment · Reviewer_vTes · 2022-08-09
> **thank you**
>
> Thanks to the authors for their detailed response. I now understand that the very limited number of condensed directions only describes the very early stage of training, and this number is expected to go up as training proceeds. I'm still not fully convinced by the implications of condensation for generalization, but I think this paper makes a good contribution towards understanding it. I have updated my score.

---

> > ### Author Response · Authors · 2022-08-10
> > **Response to Reviewer vTes**
> >
> > Dear Reviewer vTes,
> >
> > Thanks for your comments. Your comment is very helpful for our future study on the relation between condensation and generalization.
> >
> > Best,
> >
> > Authors.

---

### Meta-Review · Area_Chair_dwNu · 2022-08-27

**Recommendation:** Accept
**Confidence:** Less certain

**Metareview:**

Three reviewers recommended borderline accept, borderline accept, and weak accept. Reviewers found the work clearly written and the claims clearly stated. Of particular interest were the empirical results connecting the number of condensed directions and multiplicity of the nonlinearity, where the article was found to fill gaps and provide new intuitive explanations for important phenomena. A main critique in the initial reviews was the interest in the considered setting and the insufficient discussion of the significance of the findings, which was partly clarified during the discussion leading to updated more favorable reviewer ratings. Overall I find the strengths outweigh the weaknesses and hence I am recommending accept. However, as evidenced in the discussion, the article still can improve in some ways and I strongly encourage the authors to carefully consider the detailed feedback of the reviewers for the preparation of the final manuscript. Specific recommendations include that the revised manuscript better elucidates how the very condensed stage that the authors study is related to expressive, fully trained NNs, and includes a clearer discussion of the motivation and significance, and the promised discussion of activation functions and limitations of the theoretical analysis. Following up on the discussion in response to reviewer Mpot, I would like to suggest the authors do not use their proposed title 'informal proposition’ (or 'Theorem’ or 'Proposition’ etc.) unless the statement has a formal proof and instead use titles such as 'Conjecture’ or 'Empirical observation’ if the statement does not have a formal proof.


**Award:**

No

---

### Decision · Program_Chairs · 2022-09-14

Accept